# Alleviating the Equilibrium Challenge with Sample Virtual Labeling for Adversarial Domain Adaptation

## ABSTRACT

Numerous domain adaptive object detection (DAOD) methods leverage domain adversarial training to align the features to mitigate domain gap, where a feature extractor is trained to fool a domain classifier in order to have aligned feature distributions. The discrimination capability of the domain classifier is easy to fall into the local optimum due to the equilibrium challenge, thus cannot effectively further drive the training of feature extractor. In this work, we propose an efficient optimization strategy called Virtual-label Fooled Domain Discrimination (VFDD), which revitalizes the domain classifier during training using *virtual* sample labels. Such virtual sample label makes the separable distributions less separable, and thus leads to a more easily confused domain classifier, which in turn further drives feature alignment. Particularly, we introduce a novel concept of *virtual* label for the unaligned samples and propose the *Virtual-$\mathcal{H}$-divergence* to overcome the problem of falling into local optimum due to the equilibrium challenge. The proposed VFDD is orthogonal to most existing DAOD methods and can be used as a plug-and-play module to facilitate existing DAOD models. Theoretical insights and experimental analyses demonstrate that VFDD improves many popular baselines and also outperforms the recent unsupervised domain adaptive object detection models.

## CCS CONCEPTS

• **Computing methodologies** → **Object detection**.

## KEYWORDS

Domain Adaptation, Transfer Learning, Object Detection

**ACM Reference Format:**
Anonymous Author(s). 2018. Alleviating the Equilibrium Challenge with Sample Virtual Labeling for Adversarial Domain Adaptation. In *Proceedings of Make sure to enter the correct conference title from your rights confirmation emai (Conference acronym 'XX).* ACM, New York, NY, USA, 9 pages. https://doi.org/XXXXXXX.XXXXXXX

## 1 INTRODUCTION

Object detection has shown great success in the deep learning era [2, 3, 30, 37, 38, 53]. However, these methods assumes that source domain and target domain have the identical data distribution, which is not applicable in real-world scenarios. In addition, collecting

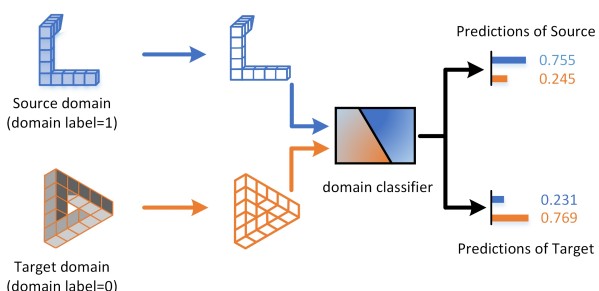

**Figure 1: Illustration of our motivation. In the case of Penrose triangle, even identical objects are challenging to ideally deceive domain classifiers (i.e., probabilities of domain classifier close to 0.5) due to variations in target angles, which in turn provides less driving power to the feature extractor for alignment and prevents effective optimization.**

large amounts of labeled samples is a time-consuming and laborious project. Unsupervised Domain Adaptation (UDA) [8, 12, 26, 34] serves as a promising solution to solve this problem by transferring knowledge from a labeled source domain to a fully unlabeled target domain.

Regarding UDA, [1] conducted theoretical analysis, demonstrating that minimizing the domain divergence between the source and target domains can effectively lower the upper bound of the target domain error. Many UDA for object detection (DAOD) methods [5, 19, 46, 48, 51] have recently attempted to learn domain invariant feature representations within de facto detection frameworks, e.g., Faster RCNN [38] and Deformable DETR [54]. Inspired by the Generative Adversarial Networks (GANs) [16], adversarial learning has been successfully applied for DAOD. The core idea behind adversarial learning methods involves training a domain classifier (i.e, $\mathcal{D}$) to distinguish between the source (i.e, $\mathcal{D}_s$, domain label=1) and target (i.e, $\mathcal{D}_t$, domain label=0) domains, while simultaneously training the feature generator (i.e, $\mathcal{G}$) to minimize the feature discrepancy between the domains, aiming to fool the discriminator in a minmax two-player game. Formally,

$$\min_{\mathcal{G}} d_{\mathcal{H}}(\mathcal{D}_s, \mathcal{D}_t) \propto \min_{\mathcal{G}} 2(1 - \min_{d \in \mathcal{D}} err(d(x)))$$
$$\propto \min_{\mathcal{G}} \max_{d \in \mathcal{D}} err(d(x))) \quad (1)$$

Where $err(d(x))$ denotes the prediction error of the domain classifier $\mathcal{D}$ and $\min_{d \in \mathcal{D}} err(d(x))$ means the minimum prediction error of an ideal domain classifier. It is worth noting that *the domain divergence $d_{\mathcal{H}}(\mathcal{D}_s, \mathcal{D}_t)$ is inversely proportional to the error rate of the domain classifier $\mathcal{D}$.* All of the existing DAOD methods generally believe that directly fooling the domain discriminator to minimize the $\mathcal{H}$-divergence will help to align the domains. However, [31]

claim that there is no guarantee that the two domains can be perfectly aligned due to the **equilibrium challenge** in adversarial learning.

Taking the Penrose Triangle as an example in Figure 1, simply due to different visual angles, the source domain may be recognized as a '⌐' while the target domain may be recognized as '△'. Consequently, the domain classifier cannot be impeccably deceived, leading to domain classification outcomes deviating significantly from the ideal equilibrium point 0.5. *Such imperfect classification by the domain classifier in turn provides less driving power for alignment in the feature extractor, impeding effective optimization*, even though there are still not aligned samples in the feature space. In other words, matching the feature distributions between domains inevitably results in sub-optimal risk due to the equilibrium challenge of adversarial learning, which makes it impossible to completely confuse the domain classifier during training. This unaligned phenomenon would be more severe in cross-domain object detection, given the complex combinations of various objects and the differentiated scene layouts between domains.

To further verify the above observations, we conduct some experiments on real cross-domain object detection scenarios. Figure. 2(a) shows one representative example of the probability distribution of instance-level features extracted by DAF [5] on the task from Cityscapes [6] (i.e., domain label=1) to Foggy Cityscapes [40] (i.e., domain label=0). Ideally, the probabilities of instance-level features obtained by adversarial alignment are evenly distributed around 0.5. However, in Figure 2(a), we can clearly observe that the predictions of source and target are distributed around 0.5260 and 0.4743 (by observing the horizontal and vertical coordinates), respectively. Therefore, a mechanism that could ideally fool the domain classifier to re-energize the adversarial training of feature generator is highly desired. This issue is rarely investigated in previous works, but it is a meaningful topic worthy of attention.

In this paper, to solve the aforementioned problem, we propose a simple but effective strategy named Virtual-sample Fooled Domain Discrimination (VFDD) which aims to fool the domain classifier to re-energize the feature generator during the training. Particularly, as illustrated in Figure 3, instead of fooling the domain classifier using real source and target samples, we propose to exploit *virtual* domain samples, where we copy the "unaligned" samples, i.e., those far from the equilibrium point 0.5, as virtual samples, in each mini-batch to be optimized/trained. The rational behind is that, for the VFDD adversarial alignment, using the same instance (i.e., *virtual* vs. real) features of the source or the target domains for alignment can naturally *maximize* the error rate of the domain classifier $\mathcal{D}$, which enables the features domain-invariant in the course of the optimization. As a result, the detector will produce more confusing instance features. One evidence is shown in Figure. 1(b), in which we implement the VFDD on DAF [5]. We can see that the probability distributions of different domains are more similar, which means that using the VFDD can better bridge the domain gap. Besides, we provide a theoretical analysis of the proposed *Virtual–$\mathcal{H}$-divergence* that it has a smaller upper-bound than the standard $\mathcal{H}$-divergence. Therefore, the effectiveness of VFDD is guaranteed. The proposed VFDD is universal and can be quickly embedded in alignment-based unsupervised domain adaptive object detection

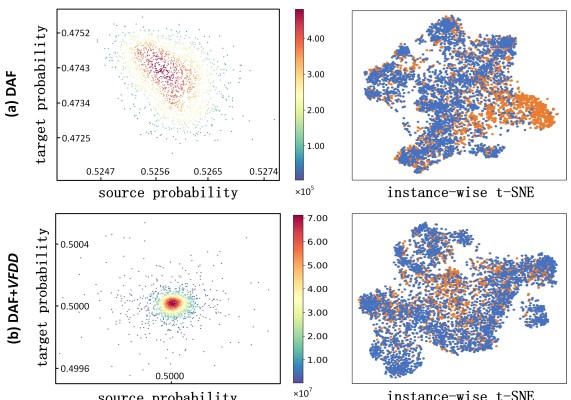

**Figure 2: Visualization of the relationship between probability distributions of domain classifier on the source and target domains and the instance-wise t-SNE of feature generator on the closed-set scenario task from cityscape to foggy cityscape. In column 1, the horizontal axis corresponds to the probability distribution of the source domain, while the vertical axis represents that of the target domain. The intensity of the color gradient in the plot reflects the density of the distributions, with brighter colors indicating denser regions. In column 2, orange signifies the source domain and blue signifies the target domain. Clearly, the probability distributions of the source and target become closer to the ideal saddle point of 0.5 and the feature distributions of the source and target domains become more consistent with ours VFDD.**

methods. The main contributions and novelties of this paper are summarized as follows:

- We pinpoint that the popular adversarial domain adaptive object detection approaches in general face optimization difficulty, which is caused by the deteriorated extraction capability of feature generator as the domain classifier falling into local optimum in training.
- We propose an efficient optimization strategy named Virtual-sample Fooled Domain Discrimination (VFDD), which is capable of re-energizing the feature generator by ideally fooling the domain classifier during training which in turn further drives feature alignment. Figure 3 is a representative and illustrative example to depict the usage of VFDD.
- Extensive experiments validate the effectiveness and universality of VFDD on various benchmark databases on several DAOD baselines. More insights and analyses of our model are also provided to justify the reasonability of VFDD and demonstrate its superiority.

## 2 RELATED WORK

**Classification Based UDA.** Unsupervised domain adaptation (UDA) is extensively studied, which aims to apply model learned from labeled source domain samples to relevant and unlabeled target domain samples [18, 33, 45]. At present, the mainstream approaches for UDA tend to realize domain alignment by adversarial learning domain-invariant feature representations across domains. Domain

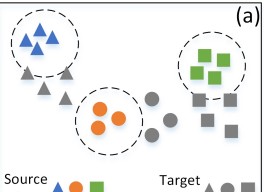 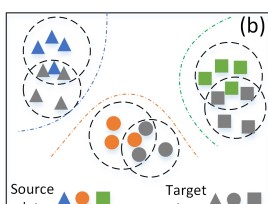 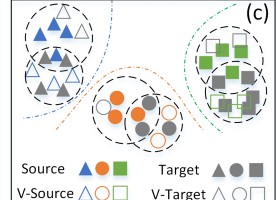 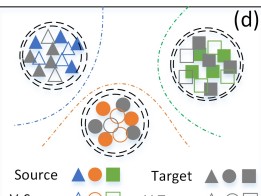 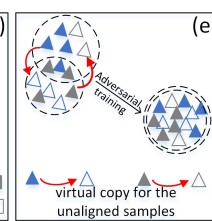

**Figure 3: An overview of the proposed Virtual-sample Fooled Domain Discrimination (VFDD) approach to domain adaptive object detection. Gray shapes are data of target domain and shapes in color are data of source domain. Different kinds of shapes indicate different classes. (a) An example of domain adaptation problem, there are source and target domains with different styles. (b) Previous methods align domain distributions by adversarially training the domain classifier with domain labels of the samples. Some samples are already aligned (i.e., the intersection of two circles) while some other samples are still not aligned, which is the well-known local optimum caused by equilibrium challenge.(c) We introduce the *virtual* source samples (i.e, V-Source) and *virtual* target (i.e., V-target) samples for the unaligned well samples to ideally fool the domain classifier to disrupt the domain classifier from being trapped in local optima, consequently revitalizing the feature extractor. (d) Depicts the state post final distribution alignment, target data are close to their source domain counterparts. (e) Illustration of introducing corresponding virtual samples for unaligned well samples. Note that the manipulation of *virtual* copy means the domain label (either real or *virtual*) is changed but without changing the feature representation of each sample. *Best viewed in color.***

Adversarial Neural Network (DANN) [14] is a representative work, where a domain classifier is connected to the feature extractor via a gradient reversal layer (GRL)[12]. Subsequently, CDAN [31] proposes an adversarial adaptation model for the discriminative information transmitted in the prediction of the classifier. GVB [9] learns the domain-invariant feature representations by applying the gradually vanishing bridge mechanism on the feature generator. DWL [50] dynamically weights the learning losses of alignment and discriminability by introducing the degree of alignment and discriminability. Nevertheless, these studies focus on the image classification and segmentation (a.k.a. pixel classification), rather than the task of object detection, which faces more challenges.

**Object Detection Based UDA.** Driven by domain adaptation theory, researchers have proposed various approaches to alleviate the problem of domain gap in cross-domain object detection. DAF [5] proposes an adaptive Faster-RCNN [38] method, which achieves image-level and instance-level feature alignment by using adversarial gradient reversal for the first time. Building upon this, MAF [19] introduces a hierarchical adversarial feature alignment strategy that reduces domain disparity at different scales. HTCN [4] employ CycleGAN for data augmentation, generating intermediate domain images to facilitate model alignment between source and target domains. VDD [49] tackles the problem by disentangling domain-invariant and domain-specific representations using vector decomposition, while also exploring the extraction of instance-invariant features [48]. IDF [28] propose a non-adversarial domain discriminator to extract domain-specific features. Additionally, PTMAF[21] and PAATF[22] introduce additional constraints during the adversarial learning stage. Recently, regarding the Transformer object detector, existing adaptation techniques for DETR [3] predominantly rely on model-based approaches SFA [47], aiming to reduce the distribution shift between different domains through sequence feature alignment.

However, as the optimization of the min-max game in domain adversarial adaptation proceeds, domain classifier is inevitably prone

to falling into local optimum due to the equilibrium challenge, which reduces the driving power for the feature alignment. In this work, we introduce the *virtual* source and *virtual* target samples to disrupt the domain classifier from being trapped in local optima, enabling the domain classifier to be perfectly fooled, thus re-energizing the feature extractor to acquire more domain-invariant features.

## 3 OUR VFDD METHOD

Following the general formula of domain adaptive object detection, we denote the labeled source domain as $\mathcal{D}_s = \{x_i^s, y_i^s\}_{i=1}^{n_s}$ with $n_s$ samples covering $C$ classes, and the unlabeled target domain as $\mathcal{D}_t = \{x_i^t\}_{i=1}^{n_t}$ with $n_t$ samples that belong to the same $C$ classes. $\mathcal{D}_s$ and $\mathcal{D}_t$ share the same feature space and category space, but have different data distributions. DAOD aims to utilize the labeled data $\mathcal{D}_s$ and unlabeled data $\mathcal{D}_t$ to learn a deep model, which can accurately predict the class label of samples in the target domain.

### 3.1 A General Framework of DAOD

Adversarial learning has proven to be an effective method for domain alignment, starting from Domain Adversarial Neural Network (DANN) [14]. The basic idea is to use the feature generator $\mathcal{G}$ to generate features to trick the domain discriminator $\mathcal{D}$. Then the domain classifier predicts whether the generated feature by $\mathcal{G}$ is from the source domain or the target domain. The training of domain alignment is achieved through the game between generator and domain classifier. The parameter $\theta_g$ of generator $\mathcal{G}$ and the parameter $\theta_d$ of domain discriminator $\mathcal{D}$ are optimized by the following domain alignment objective function.

$$\mathcal{L}_{adv}(\theta_g, \theta_d) = \mathbb{E}_{x_i^s \sim \mathcal{D}_s} \log \left[ \mathcal{D}(\mathcal{G}(x_i^s)) \right] \\ + \mathbb{E}_{x_i^t \sim \mathcal{D}_t} \log \left[ 1 - \mathcal{D}(\mathcal{G}(x_i^s)) \right] \quad (2)$$

In order to improve the classification performance of the target domain samples, we must first ensure that the classifier $C$ can

correctly classify the samples from the source domain. Thus, the supervised classification loss can be described as:

$$\mathcal{L}_{cls}(\theta_g, \theta_c) = \frac{1}{n_s} \sum_{i=1}^{n_s} \mathcal{L}_{ce}(C(\mathcal{G}(x_i^s; \theta_g); \theta_c), y_i^s) \quad (3)$$

where $\mathcal{L}_{ce}$ is the standard cross-entropy loss function. During training stage, the existing methods usually jointly optimize the two objective functions ($\mathcal{L}_{dom}$ and $\mathcal{L}_{cls}$). The overall minimax objective function is:

$$\min_{\theta_g, \theta_c} \max_{\theta_d} \mathcal{L}_{cls} + \eta \mathcal{L}_{adv} \quad (4)$$

where $\theta_g$, $\theta_d$, $\theta_c$ denote the parameters of feature generator, domain discriminator and category classifier, respectively. $\eta$ denotes a hyper-parameter for balancing the losses.

## 3.2 Theoretical Motivation of VFDD

During the training, the discrimination capability of the domain classifier is easy to fall into the local optimum due to the equilibrium challenge. This in turn negatively influences the effectiveness of the optimization towards feature alignment. Thus, with the adversarial optimization objectives as in Eq.4, the networks converge to states that are less than optimal. Even though the distributions are not fully aligned, the deteriorated discrimination capability of the domain classifier cannot further efficiently drive the feature alignment.

We introduce the *virtual* source and *virtual* target samples for the unaligned samples to perfectly fool the domain classifier to re-energize the feature extractor to acquire more domain-invariant features, which then formulates the proposed VFDD technique. As described in Eq.1, the domain distance between source and target is inversely proportional to the error rate of the domain classifier $\mathcal{D}$. Intuitively, larger domain prediction error means smaller domain discrepancy. Thus, in order to fully fool the domain, we propose the *Virtual-$\mathcal{H}$*-divergence which aims at minimizing the domain discrepancy without information loss. Instead of directly calculating the distance between $\mathcal{D}_s$ and $\mathcal{D}_t$ via $\mathcal{H}$-divergence (i.e., $d_{\mathcal{H}}(\mathcal{D}_s, \mathcal{D}_t)$ in Eq.1 ), we introduce the concept of *virtual* domain samples and learn domain invariant features between *virtual* and real domain to implicitly align the features across source and target domains. Specifically, we introduce the *virtual* source domain $\hat{\mathcal{D}}_s$ with domain label $d = 0$ instead of $d = 1$ and the *virtual* target domain $\hat{\mathcal{D}}_t$ with domain label $d = 1$ instead of $d = 0$, by simply copying the feature matrices of real source $\mathcal{D}_s$ ($d = 1$) and real target $\mathcal{D}_t$ ($d = 0$) for easier manipulation. We propose to measure the distances between $\mathcal{D}_s$ and $\hat{\mathcal{D}}_s$ and the distances between the $\mathcal{D}_t$ and $\hat{\mathcal{D}}_t$ by:

$$d_{\mathcal{H}}(\mathcal{S}, \hat{\mathcal{S}}) = 2[1 - \min(err(\mathcal{D}(x_s)) + err(\mathcal{D}(\hat{x_s})))]$$
$$d_{\mathcal{H}}(\hat{\mathcal{T}}, \mathcal{T}) = 2[1 - \min(err(\mathcal{D}(\hat{x_t})) + err(\mathcal{D}(x_t)))] \quad (5)$$

Due to the $err(\cdot)$ refers to the cross entropy loss, the *Virtual-$\mathcal{H}$*-divergence can be written as:

$$d_{V-\mathcal{H}}(\mathcal{S}, \mathcal{T}) = \frac{1}{2}[d_{\mathcal{H}}(\mathcal{S}, \hat{\mathcal{S}}) + d_{\mathcal{H}}(\hat{\mathcal{T}}, \mathcal{T})]$$
$$= 2 - \min\left(err(\mathcal{D}(\hat{x_s})) + err(\mathcal{D}(x_s))\right.$$
$$\left. + err(\mathcal{D}(\hat{x_t})) + err(\mathcal{D}(x_t))\right)$$
$$= 2\left[1 - \min\left(err(\mathcal{D}(x_s \oplus \hat{x_t}))\right.\right.$$
$$\left.\left. + err(\mathcal{D}(\hat{x_s} \oplus x_t))\right)\right]$$
$$= d_{\mathcal{H}}(\mathcal{S} \oplus \hat{\mathcal{T}}, \hat{\mathcal{S}} \oplus \mathcal{T}) \quad (6)$$

where $\oplus$ denotes the *union* operator. Because the features of the ($\mathcal{S} \oplus \hat{\mathcal{T}}$) and ($\hat{\mathcal{S}} \oplus \mathcal{T}$) are identical except for the domain labels, the error rate of the domain classifier $\mathcal{D}$ is very large. This means the value of $d_{\mathcal{H}}(\mathcal{S} \oplus \hat{\mathcal{T}}, \hat{\mathcal{S}} \oplus \mathcal{T})$ is much smaller than $d_{\mathcal{H}}(\mathcal{S} \oplus \mathcal{S}, \mathcal{T} \oplus \mathcal{T})$. Formally,

$$d_{V-\mathcal{H}}(\mathcal{S}, \mathcal{T}) = d_{\mathcal{H}}(\mathcal{S} \oplus \hat{\mathcal{T}}, \hat{\mathcal{S}} \oplus \mathcal{T})$$
$$\leq d_{\mathcal{H}}(\mathcal{S} \oplus \mathcal{S}, \mathcal{T} \oplus \mathcal{T}) = d_{\mathcal{H}}(\mathcal{S}, \mathcal{T}) \quad (7)$$

In this way, *such virtual sample makes the separable distributions less separable, and thus leads to a more easily confused domain classifier, which in turn further drives feature alignment*. As revealed in Figure 2 (a) and (b), after applying VFDD, the domain-invariant information extraction capability of the feature extractor is stronger than the baseline.

Based on the above novel *Virtual-$\mathcal{H}$*-divergence, the expected error on target samples $\epsilon_{\mathcal{T}}$ is bounded as,

$$\epsilon_{\mathcal{T}} \leq \epsilon_{\mathcal{S}} + \frac{1}{2}d_{V-\mathcal{H}}(\mathcal{S}, \mathcal{T}) + \lambda$$
$$\leq \epsilon_{\mathcal{S}} + \frac{1}{2}d_{\mathcal{H}}(\mathcal{S}, \mathcal{T}) + \lambda \quad (8)$$

where $\epsilon_{\mathcal{S}}$ is the expected error on the source domain, $\lambda$ is the ideal joint hypothesis. In this way, the upper bound of the expected target error, *i.e.*, $\epsilon_{\mathcal{T}}$ can be effectively reduced in our work. We show the derivation of Inequality 8 in the **supplementary** material.

## 3.3 Proposed Training Strategy VFDD+DAOD

With the general DAOD (i.e., Eq. 4) described in Section 3.1 and the VFDD based *Virtual-$\mathcal{H}$*-divergence strategy in Eq. 6, a balanced DAOD model can be trained and implemented.

**Measurement of Alignment for a Sample.** To virtual copy the "unaligned" samples (i.e., as shown in Figure 3 (e)), we need to evaluate whether a sample is "unaligned" or not. As we know, when a sample is more aligned, it is harder for the domain classifier to identify its domain and has a higher uncertainty w.r.t. to the predicted domain label of this sample. Correspondingly, the entropy of domain classification of this sample is in general higher. Therefore, we measure how well a target sample is aligned with the source domain by simply using the entropy of domain classification,

$$H(p) = -p\log(p) - (1 - p)\log(1 - p) \quad (9)$$

where $p$ denotes the probability of predicting a sample $x$, i.e., $p = \mathcal{D}(\mathcal{G}(x))$. The larger the entropy value, the larger the ambiguity/uncertainty of the discriminator has when identifying which domain it belongs to, and the more aligned the sample is. Note that there may exist other more accurate or advanced measurement metrics, but this is not the focus of this paper and we use this simple one here.

**Unaligned Sample Selection and Virtual Copy.** For a given sample, when it can be well distinguished by the domain classifier, it could be considered approximately unaligned. In other words, when the entropy of the domain classifier prediction is smaller than a threshould $\tau$, we define it as a "unaligned" sample and introduce a corresponding *virtual* sample (as shown in Figure 3 (e)). $\tau$ is a hyper-parameter that controls the strictness of the definition of "unalign". We will study its influence in the experiments.

**When to start VFDD.** Intuitively, we could start our *virtual* copy strategy to fool the domain classifier whenever its discrimination capability begins to fall into local optimum. In general, at the early training stage, the optimization of the domain discriminator persistently improves its discrimination capability. It is thus unnecessary to enable the *virtual* copy strategy. Inspired by the popular learning rate (lr) adjustment algorithms [36] which adjust the learning rate if no improvement is seen for a "patience" number of epochs (where "patience" is usually set to 2 to 10) , we start VFDD if no improvement of the discrimination capability is seen for a "patience" number of epochs and we denote this hyper-parameter as $K$.

**Adversarial Training.** As is done in previous adversarial domain adaptation methods, we perform mini-batch level optimization where a batch consists of both source and target domain samples. Once VFDD is activated, for each mini-batch, we first check whether each sample should be copied as a *virtual* sample and copy them if deemed so. Then the adversarial training is performed between the *virtual* and real domain labels. The novel domain alignment objective function for the unaligned samples can be expressed as:

$$
\begin{aligned}
\mathcal{L}_{adv}^{V}(\theta_g, \theta_d) =& \mathbb{E}_{x_i^s \sim \mathcal{D}_s} \log\left[\mathcal{D}(\mathcal{G}(x_i^s))\right] \\
&+ \mathbb{E}_{x_i^t \sim \mathcal{D}_t} \log\left[1 - \mathcal{D}(\mathcal{G}(x_i^t))\right] \\
&+ \mathbb{E}_{\hat{x}_i^t \sim \hat{\mathcal{D}}_t} \log\left[\mathcal{D}(\mathcal{G}(\hat{x}_i^t))\right] \\
&+ \mathbb{E}_{\hat{x}_i^s \sim \hat{\mathcal{D}}_s} \log\left[1 - \mathcal{D}(\mathcal{G}(\hat{x}_i^s))\right]
\end{aligned}
\tag{10}
$$

Thus, the domain alignment objective function for the all source and target samples can be expressed as:

$$
\mathcal{L}_{adv}^{all}(\theta_g, \theta_d) =
\begin{cases}
\mathcal{L}_{adv}(\theta_g, \theta_d) & H(p) \geq \tau \\
\mathcal{L}_{adv}^{V}(\theta_g, \theta_d) & H(p) \leq \tau
\end{cases}
\tag{11}
$$

where $\mathcal{L}_{adv}(\theta_g, \theta_d)$ and $\mathcal{L}_{adv}^{V}(\theta_g, \theta_d)$ are calculated by the Eq.2 and Eq.10, respectively. The domain classifier is updated by maximizing the domain classification loss (i.e., adversarial loss as in Eq.11) over the updated source set and target set. Simultaneously, the feature extractor is trained to acquire more domain-invariant (confused) features. Note that the *virtual* copy strategy has no impact on object classification loss.

With the thoughts above, the overall training objective is:

$$
\min_{\theta_g, \theta_c} \max_{\theta_d} \mathcal{L}_{cls} + \mathcal{L}_{adv}^{all}
\tag{12}
$$

The pseudo code is presented in the Algorithm 1, showing the training process of VDFF+DAOD. This endeavor will contribute to a more precise assessment of our research. The code will be made publicly available.

---

**Algorithm 1** VFDD+DAOD Optimization Algorithm

---

**Require:** A source set $\mathcal{D}_s = \{x_i^s, y_i^s\}_i^{n_s}$, including the images and labels (i.e., bounding box coordinates and category labels). An unlabeled target training set $\mathcal{D}_t = \{x_i^t\}_i^{n_t}$.

**Ensure:** A detector adaptive to different domains.

1: **Initialization.** The iteration counter $n$, the total iteration number $m$.

2: Load the parameters of the pre-trained model (i.e., VGG-16 or ResNet-50) to the VFDD+DAOD.

3: **Repeat:**

4:     Take the samples from the source set $\mathcal{D}_s$, including the image $x_i^s$ and label $y_i^s$. Take the samples $x_i^t$ from the target set $\mathcal{D}_t$.

5:     Feed the source sample and corresponding labels into the VFDD+DAOD part, and compute $\mathcal{L}_{cls}$ in Eq.(3) in our manuscript.

6:     Feed the source and target samples into the VFDD+DAOD part. Unaligned samples selection and virtual copy with the threshold $\tau$, then compute the adversarial learning loss $\mathcal{L}_{adv}^{all}$ in Eq.(11) in our manuscript.

7:     Compute the total loss Eq.(12) and gradient, and update the parameters of the VFDD+DAOD part.

8:     $n = n + 1$

9: **until** $n = m$

---

## 4 EXPERIMENTS AND RESULTS

### 4.1 Experimental Setup

**Datasets and Settings.** We test our method on Cityscapes [7], Foggy Cityscapes [41], Sim10k [25], and BDD100k [52]. These datasets cover diverse and challenging scenarios for domain adaptation tasks:

- **Weather Adaptation.** In this scenario, we use Cityscapes as the source dataset, consisting of 2,975 training images and 500 evaluation images. The counterpart, known as Foggy Cityscapes, is derived from Cityscapes through a fog synthesis algorithm. These datasets enable us to assess the effectiveness of our method in adapting object detection models from clear weather to foggy conditions.

- **Synthetic to Real Adaptation.** In this particular scenario, we leverage Sim10k as the source domain, generated using the Grand Theft Auto game engine. Sim10k includes 10,000 training images with 58,701 bounding box annotations. For the target domain, we utilize car instances from Cityscapes for training and evaluation.

- **Scene Adaptation.** In this condition, Cityscapes functions as the source dataset, while the target dataset is the daytime subset of BDD100k. The BDD100k subset comprises 36,728 training images and 5,258 validation images, all meticulously annotated with bounding boxes. This subset encompasses a wide array of scenes captured during daylight hours.

**Table 1: Results of different methods for weather adaptation, *i.e.*, Cityscapes to Foggy Cityscapes. FRCNN and DefDETR are abbreviations for Faster RCNN based on the VGG-16 and Deformable DETR based on the ResNet-50, respectively.**

| Methods | Detector | person | rider | car | truck | bus | train | mcycle | bicycle | mAP |
|---------|----------|--------|-------|-----|-------|-----|-------|--------|---------|-----|
| FRCNN | FRCNN | 24.1 | 33.1 | 34.3 | 4.1 | 22.3 | 3.0 | 15.3 | 26.5 | 20.3 |
| DAF | FRCNN | 25.0 | 31.0 | 40.5 | 22.1 | 35.3 | 20.2 | 20.0 | 27.1 | 27.6 |
| MAF | FRCNN | 28.2 | 39.5 | 43.9 | 23.8 | 39.9 | 33.3 | 29.2 | 33.9 | 34.0 |
| ATF | FRCNN | 34.6 | 48.0 | 50.0 | 23.7 | 43.3 | 38.7 | 33.4 | 38.8 | 38.7 |
| HTCN | FRCNN | 33.2 | 47.5 | 47.9 | 31.6 | 47.4 | 40.9 | 32.3 | 37.1 | 39.8 |
| UMT | FRCNN | 33.0 | 45.9 | 48.6 | 34.1 | 56.5 | 46.8 | 30.4 | 37.3 | 41.7 |
| PAATF | FRCNN | 37.9 | 49.6 | 52.8 | 27.0 | 46.6 | 48.7 | 33.6 | 39.5 | 42.0 |
| PTMAF | FRCNN | 37.3 | 49.4 | 52.2 | 26.7 | 49.5 | 34.5 | 34.9 | 41.2 | 40.7 |
| IDF | FRCNN | 37.4 | 50.1 | 52.8 | 31.3 | 50.6 | 42.0 | 33.7 | 41.7 | 42.4 |
| DAF+**VFDD** | FRCNN | 31.4(+6.4) | 40.8(+9.8) | 43.3(+2.8) | 16.4(-5.7) | 38.7(+3.4) | 27.6(+7.4) | 23.5(+3.5) | 33.2(+6.1) | 31.9(+4.3) |
| MAF+**VFDD** | FRCNN | 30.6(+2.4) | 41.7(+2.2) | 46.1(+2.2) | 24.7(+0.9) | 42.0(+2.1) | 43.1(+9.8) | 30.7(+1.5) | 35.5(+1.6) | 36.8(+2.8) |
| IDF+**VFDD** | FRCNN | 38.2(+0.8) | 51.3(+1.2) | 54.4(+1.6) | 33.1(+1.8) | 50.7(+0.1) | 43.3(+1.3) | 34.8(+1.1) | 42.3(+0.6) | **43.5**(+1.1) |
| DefDETR | DefDETR | 37.7 | 39.1 | 44.2 | 17.2 | 26.8 | 5.8 | 21.6 | 35.5 | 28.5 |
| SFA | DefDETR | 46.5 | 48.6 | 62.6 | 25.1 | 46.2 | 29.4 | 28.3 | 44.0 | 41.3 |
| SFA+**VFDD** | DefDETR | 47.2(+0.7) | 50.1(+1.5) | 65.5(+2.9) | 25.7(+0.6) | 47.3(+1.1) | 31.1(+1.7) | 31.7(+3.4) | 45.7(+1.7) | **43.0**(+1.7) |

**Implementation Details.** We benchmark our approach against cutting-edge domain adaptation methods across two categories: (1) The Faster RCNN series, including DAF [5], MAF [19], ATF [20], HTCN [4], UMT [10], PAATF [22], PTMAF [21], and IDF [28]. (2) The Deformable DETR series, encompassing SFA [47]. In addition, we also compare our VFDD with the image classification task, and more details will be presented in the **Supplementary**.

We validate the effectiveness and versatility of our method against various baselines, including **DAF**, **MAF**, **IDF**, and **SFA**. By default, we employ ImageNet pre-trained ResNet-50 [17] and VGG-16 [42] as CNN backbones in all experiments. Aligning with the Faster RCNN series, we train the network using the SGD optimizer with a momentum of 0.9 and a weight decay of $5 \times 10^{-4}$. The initial learning rate is set to $1 \times 10^{-3}$ and is reduced to $1 \times 10^{-4}$ after 5 epochs. A total of 15 epochs are conducted, with a batch size of 2 maintained throughout. In line with the Deformable DETR series, we utilize the Adam optimizer [27] for training over 50 epochs. The learning rate is initialized at $2 \times 10^{-4}$ and reduced by a factor of 0.1 after 40 epochs. A batch size of 4 is employed consistently in all experiments. All these experiments are conducted using NVIDIA Tesla V100 GPUs.

## 4.2 Comparisons with SOTA Methods

**Weather Adaptation.** To assess the reliability of object detectors under varying weather conditions, we conduct cross-domain transfers of models from Cityscapes to Foggy Cityscapes. The outcomes are presented in Table 1. Notably, the application of VFDD yields marked improvements in mAP for various methods. Specifically, DAF, MAF, IDF, and SFA achieve mAP values of 31.9%, 36.8%, 43.5%, and 43.0%, respectively, after VFDD integration. Furthermore, the results demonstrate that VFDD notably amplifies the cross-domain performance of Deformable DETR, showcasing a substantial absolute mAP gain of 14.5% (28.5% vs. 43.0%). These promising results highlight the ability of our method to improved performance in unsupervised domain adaptation for object detection tasks.

**Table 2: Results of different methods for synthetic to real adaptation, *i.e.*, Sim10k to Cityscapes.**

| Methods | Detector | car AP |
|---------|----------|--------|
| FRCNN | FRCNN | 34.6 |
| DAF | FRCNN | 38.9 |
| MAF | FRCNN | 41.1 |
| ATF | FRCNN | 42.8 |
| HTCN | FRCNN | 42.5 |
| UMT | FRCNN | 43.1 |
| PAATF | FRCNN | 43.7 |
| PTMAF | FRCNN | 43.2 |
| IDF | FRCNN | 43.9 |
| DAF+**VFDD** | FRCNN | 41.3(+2.4) |
| MAF+**VFDD** | FRCNN | 43.0(+1.9) |
| IDF+**VFDD** | FRCNN | **45.1**(+1.2) |
| DefDETR | DefDETR | 47.4 |
| SFA | DefDETR | 52.6 |
| SFA+**VFDD** | DefDETR | **54.7**(+2.1) |

**Synthetic to Real Adaptation.** Leveraging economical yet precise simulation datasets has proven to elevate the performance of object detectors. Nevertheless, this approach introduces a significant challenge in the form of a substantial inter-domain gap. In the context of synthetic-to-real adaptation, we have evaluated the efficacy of VFDD, our proposed method, as outlined in Table 2. We can see that our VFDD significantly outperforms the baseline DAF, MAF, IDF, and SFA by 2.4%, 1.9%,1.2% and 2.1%,respectively.

**Scene Adaptation.** In real-world applications, such as autonomous driving, scene layouts are dynamic and subject to frequent changes. Consequently, model adaptability to scene variations becomes pivotal. VFDD, our proposed method, underscores its effectiveness in scene adaptation, as evidenced in Table 3, where it achieves state-of-the-art results (SFA+VFDD 31.4%). Notably, performance improvements are observed across all categories.

**Table 3: Results of different methods for scene adaptation,*i.e.,* Cityscapes to BDD100k daytime subset.**

| Methods | Detector | person | rider | car | truck | bus | mcycle | bicycle | mAP |
|---|---|---|---|---|---|---|---|---|---|
| FRCNN | FRCNN | 29.3 | 28.2 | 45.7 | 15.5 | 16.6 | 16.0 | 22.1 | 24.8 |
| DAF | FRCNN | 26.9 | 22.1 | 44.7 | 17.4 | 16.7 | 17.1 | 18.8 | 23.4 |
| SWDA | FRCNN | 30.2 | 29.5 | 45.7 | 15.2 | 18.4 | 17.1 | 21.2 | 25.3 |
| DAF+**VFDD** | FRCNN | 30.4(+3.5) | 29.7(+7.6) | 46.0(+1.3) | 18.1(+0.7) | 18.9(+2.2) | 17.9(+0.8) | 22.7(+3.9) | **26.2**(+2.8) |
| DefDETR | DefDETR | 38.9 | 26.7 | 55.2 | 15.7 | 19.7 | 10.8 | 16.2 | 26.2 |
| SFA | DefDETR | 40.2 | 27.6 | 57.5 | 19.1 | 23.4 | 15.4 | 19.2 | 28.9 |
| SFA+**VFDD** | DefDETR | 42.7(+2.5) | 30.1(+2.5) | 59.2(+1.7) | 22.9(+3.8) | 25.1(+1.7) | 16.7(+1.3) | 23.0(+3.8) | **31.4**(+2.5) |

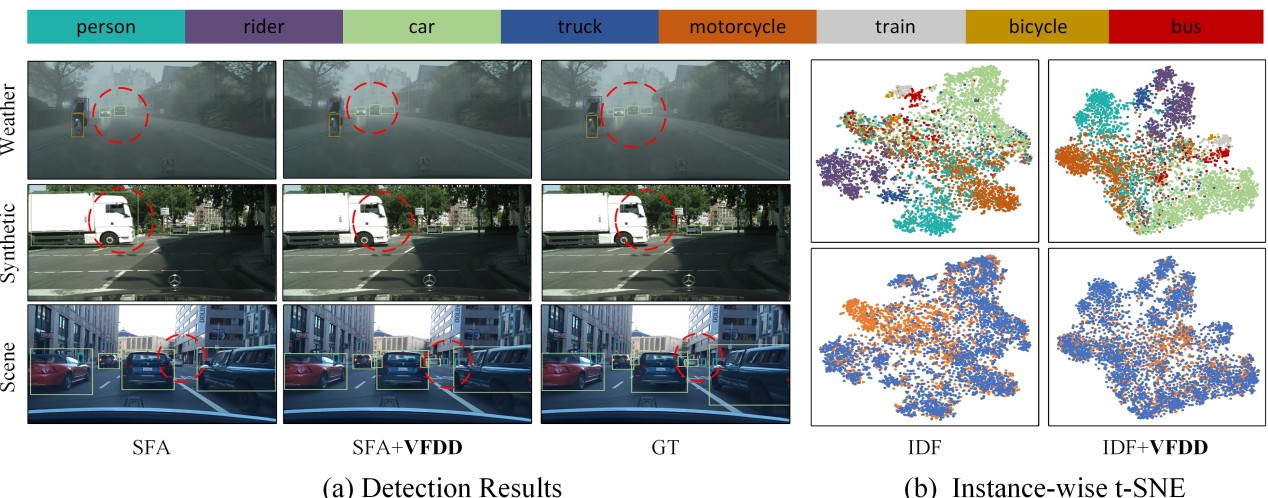

          (a) Detection Results               (b) Instance-wise t-SNE

**Figure 4: (a) Qualitative comparison of SFA+VFDD with previous SOTA method and GT in three scenarios. The red circle area reflects the superiority of our method. (b) In Cityscapes to Foggy Cityscapes, instance-level feature t-SNE results. Colors in the first row represent classes, while orange signifies the source domain and blue signifies the target domain in second row.**

## 4.3 Visualization and Analysis

**Detection Results.** We present visualizations of *SFA+***VFDD** outcomes on three target domain datasets: Foggy Cityscapes, Cityscapes, and BDD100k. These visuals are accompanied by ground truth and previous SOTA methods. In Figure 4 (a), Row 1 (Cityscapes to Foggy Cityscapes), SFA+**VFDD** exhibits enhanced recall and more accurate classification in scenarios with dense fog occlusion. In Row 2 (Sim10k to Cityscapes), our approach even mitigates label misalignment (car vs. truck) to a certain extent without supervision. In Row 3 (Cityscapes to BDD100k), our method effectively classifies and locates objects, even under heavy occlusion or challenging small sizes. These visuals match quantitative findings, confirming VFDD's effectiveness in mitigating domain shifts within the UDA Transformer detector.

**t-SNE Distribution Results.** Figure 4 (b) describes the t-SNE [44] visualizations of features learned by IDF (baseline) and IDF+**VFDD** on the weather adaptation. The visualization feature distributions employed VFDD have better clustering effect and have fewer samples distributed across class boundaries, which intuitively boosts the feature discriminability. In addition, visualization results further validate the learning ability of our VFDD mechanism. See **Supplementary** for more results.

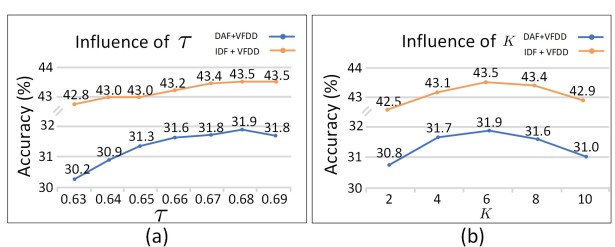

          (a)               (b)

**Figure 5: Influence of (a) threshold $\tau$, and (b) $K$.**

## 4.4 Design Choices

For clear analysis, we do experiments on our scheme VFDD for design choices study on the weather adaptation.

**Influence of Threshold $\tau$.** As described in Section 3.3, we employ a hyper-parameter $\tau$ as the threshold to determine whether a sample is "unaligned". We study its influence in Figure 5 (a). We can see that a superior performance is achieved when $\tau$ ranges from 0.63 to 0.69 on two schemes. Similar trends are observed on other datasets. We set $\tau = 0.68$ on all datasets.

**Influence of $K$.** As described in Section 3.3, we start our VFDD if the domain classifier fall into the local optimum. Figure 5 (b)

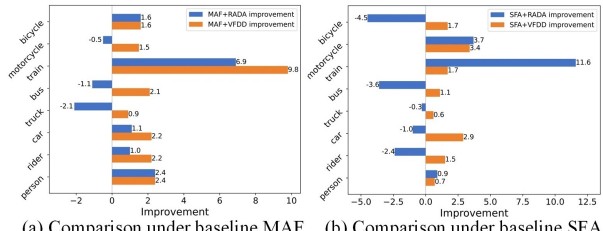

(a) Comparison under baseline MAF    (b) Comparison under baseline SFA

**Figure 6: Comparison of performance improvement of RADA and VFDD under MAF (a) and SFA (b) benchmarks in cityscape to foggy cityscape scenarios.**

illustrates the impact of $K$. We find optimal performance when $K$ falls between 4 and 6. If $K$ is too small, assessing improvement becomes unreliable due to noise sensitivity. Conversely, if $K$ is too large, the optimization strategy's full potential cannot be harnessed.

## 4.5 Discussion

**Comparison with RADA** RADA [24] utilizes dynamic domain label adjustments for the 'well-aligned' samples to re-energize the domain classifier. Conversely, our VFDD uses a unique strategy by giving 'unaligned' samples virtual labels to deceive the domain classifier. This encourages the feature generator to produce features that are indistinguishable between domains, leading to better alignment. As depicted in Figure 6, VFDD outperforms various baseline methods, exhibiting positive outcomes across diverse categories. Conversely, RADA occasionally yields negative outcomes. This unequivocally demonstrates VFDD's superiority in enhancing learning and adaptation across domains compared to RADA.

**Experimental Results on Classification** To showcase the effectiveness of VFDD, we extend our experimentation to the image classification datasets of Digits, as presented in Table 4. The table reveals that, under the optimization coordination of VFDD, the classification accuracy of MCD has reached 96.3%, 96.5%, and 95.1% on tasks $M \to U$, $U \to M$, and $S \to M$ respectively. This demonstrates that VFDD is a powerful auxiliary tool to help to achieve better object Classification performance. Further, we can observe that our method VFDD can be easily plugged and played in the existing alignment-based UDA methods to enhance their recognition performance on image classification task.

**Convergence Analysis** We present the convergence curves of test accuracy with respect to the number of iterations on tasks of $USPS \to MNIST$ as shown in Figure 7 (a). The blue line represents the test error of baselines MCD, and the red line represents the test accuracy of MCD+VFDD. Obviously, compared with baseline, MCD+VFDD has a faster convergence speed and higher test accuracy. This fully illustrates that VFDD plays an active coordination role in practical optimization process, which promotes the development of domain alignment task and classification task towards a benign direction.

**Alignment Analysis** We present the convergence curves of Maximum Mean Discrepancy (MMD) concerning the number of iterations on the $MNIST \to USPS$ tasks, as illustrated in Figure 7 (b). The blue line represents the alignment state of the baseline method MCD, while the red line represents the alignment state of

**Table 4: Performance (%) comparisons with the previous UDA approaches on Digits Datasets( MNIST(M), USPS (U) and SVHN (S)). All experiments are conducted based on ResNet-50 pre-trained on ImageNet.**

| Methods | $M \to U$ | $U \to M$ | $S \to M$ | Average |
|---|---|---|---|---|
| DANN[13] | 80.3 | 77.8 | 73.5 | 77.2 |
| DRCN [15] | 91.8 | 73.7 | 82.0 | 82.5 |
| CoGAN [29] | 91.2 | 89.1 | - | - |
| ADDA [43] | 89.4 | 90.1 | 76.0 | 85.2 |
| CAT[11] | 90.6 | 80.9 | 98.1 | 89.9 |
| TPN [35] | 92.1 | 94.1 | 93.0 | 93.1 |
| CDAN [32] | 93.9 | 96.9 | 88.5 | 93.1 |
| CyCADA [23] | 95.6 | 96.5 | 90.4 | 94.2 |
| MCD [39] | 94.2 | 94.1 | 94.5 | 94.3 |
| CDAN(baseline) | 93.9 | 96.9 | 88.5 | 93.1 |
| CDAN+**VFDD** | 94.4(+0.5) | 97.5(+0.6) | 89.5(+1.0) | 93.8(+0.7) |
| MCD(baseline) | 94.2 | 94.1 | 94.5 | 94.3 |
| MCD+**VFDD** | 96.3(+2.1) | 96.5(+2.4) | 95.1(+0.6) | **96.0**(+1.7) |

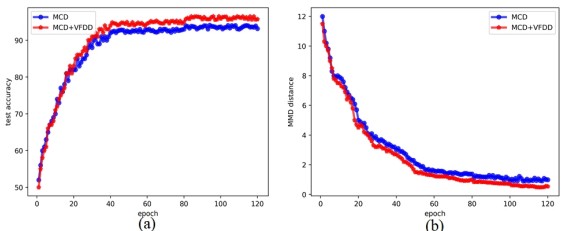

(a)    (b)

**Figure 7: (a) Convergence curves of MCD and MCD+VFDD on the test accuracy (%). (b) Alignment state measured by MMD.**

MCD combined with VFDD. Clearly, as training progresses, the MMD value gradually decreases, indicating a diminishing disparity between the source and target domains. Under the influence of VFDD, the MMD value is further diminished compared to the baseline, highlighting the efficacy of VFDD in narrowing the gap between domains. This outcome also validates the effectiveness of our proposed *Virtual-$\mathcal{H}$-divergence* (i.e., Eq. 6).

## 5 CONCLUSION

In this paper, we propose an effective optimization strategy for adversarial domain adaptation, termed as Virtual-sample Fooled Domain Discrimination (VFDD), which aims to prevent the domain classifier from getting stuck in local optima and simultaneously enhance feature alignment. We achieve this by copying the "unaligned" samples as *virtual* domain samples, which encourages the exploration of *virtual* domain labels, disrupting potential local optima entrapment of the domain classifier, and revitalizing the feature extractor in a dynamic manner. Extensive experiments across diverse benchmarks and various networks have consistently showcased the potency and broad applicability of our VFDD strategy. These results underscore its effectiveness in achieving robust and generalized adversarial domain adaptation. We aim to inspire further research to infuse greater dynamism into adversarial domain adaptation, thereby enhancing training effectiveness.

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

Received 20 February 2007; revised 12 March 2009; accepted 5 June 2009

