# OpenReview forum: "Alleviating the Equilibrium Challenge with Sample Virtual Labeling for Adversarial Domain Adaptation"
_acmmm.org/ACMMM/2024/Conference — MM2024 Poster_

### Official Review · Reviewer_htua · 2024-05-24

**Rating:** 3
**Confidence:** 4

**Summary:**

In domain adaptive object detection, traditional methods often struggle with domain gaps due to the limited discrimination capability of domain classifiers, which tend to reach a local optimum. To overcome this, the proposed Virtual-label Fooled Domain Discrimination (VFDD) strategy introduces an innovative approach using virtual sample labels to rejuvenate the domain classifier's effectiveness during training. This makes the distributions less separable, enhancing the classifier's confusion and promoting better feature alignment. VFDD, which is compatible with most existing DAOD methods as a plug-and-play module, has been shown to enhance baseline models and outperform recent unsupervised domain adaptive object detection models through theoretical insights and experimental validation.

**Strengths:**

1. The authors' experiments are good. They have demonstrated their method's effectiveness in various scenarios, such as Synthetic to Real Adaptation and weather adaptation. The authors' comparisons using different cases are decent.

2. Several motivations/claims are justified using suitable experiments/visualizations (like Fig 2).

3. The proposed methodology also betters the SOTA concerning difficult
   situations like heavy occlusion and small size of objects, thus enhancing
   the applicability of the methodology.

4. Overall, the paper is well-written and easy to follow.

**Limitations:**

The manuscript's explanation of how the Virtual-label Fooled Domain Discrimination (VFDD) enhances feature alignment lacks clarity, which is crucial for understanding the efficacy of this method in domain adaptation challenges. A detailed step-by-step breakdown of the VFDD mechanism should be provided to improve the manuscript, explicitly showing how virtual labels contribute to improved feature alignment across domains.

Furthermore, the study's selection of baselines for comparison appears to rely on somewhat dated methodologies. In a rapidly evolving field like machine learning, it is imperative to accurately compare new techniques against the most recent and relevant baselines to establish their effectiveness and innovation. The authors should consider incorporating more current state-of-the-art methods that utilize similar or competing approaches to domain adaptation.

An F-divergence analysis would add depth to the evaluation of VFDD, offering insights into how different the distributions are before and after applying VFDD.

Comparing the performance of a Vision Transformer (ViT)--based backbone with that of other architectures like CNNs in the VFDD setup could offer significant insights. Since ViTs are known for their efficacy in handling different types of data distributions due to their attention mechanisms, evaluating VFDD with a ViT backbone could reveal additional benefits or limitations of the approach.

Lastly, providing qualitative analysis concerning the overlap of the domains post-application of VFDD would be beneficial. This could include visualizations of feature spaces before and after applying VFDD.

**Suitability:**

2

---

### Official Review · Reviewer_WztL · 2024-05-25

**Rating:** 4
**Confidence:** 3

**Summary:**

This paper addresses the equilibrium challenge in Domain Adaptive Object Detection (DAOD), where the domain classifier's discrimination capability degrades over time, hindering effective feature alignment. To solve this, the authors propose a novel optimization strategy called Virtual Label Fooled Domain Discrimination (VFDD), which uses virtual sample labels to rejuvenate the domain classifier, facilitating better feature alignment. This method introduces virtual labels for "unaligned" samples to confuse the domain classifier and improve feature alignment by making the distributions less separable. VFDD is effective when integrated with existing DAOD methods, as demonstrated through theoretical insights and extensive experiments on benchmarks such as Cityscapes, Foggy Cityscapes, Sim10k, and BDD100k. The experimental results show significant improvements in cross-domain object detection tasks, and the method's effectiveness is further validated through t-SNE plots and qualitative results. VFDD's ability to enhance feature alignment and domain adaptation makes it a valuable addition to current DAOD techniques.

**Strengths:**

The paper addresses the equilibrium challenge in Domain Adaptive Object Detection (DAOD) where the discrimination capability of the domain classifier degrades, hindering effective feature alignment. The authors propose a novel optimization strategy called Virtual Label Fooled Domain Discrimination (VFDD), which uses virtual sample labels to rejuvenate the domain classifier, facilitating better feature alignment. VFDD can be integrated with existing DAOD methods to enhance their performance.
1.	The paper presents an innovative optimization strategy, Virtual Label Fooled Domain Discrimination (VFDD), to tackle the equilibrium challenge in adversarial domain adaptation. This novel approach of using virtual sample labels to confuse the domain classifier is a good contribution to the field.
2.	The authors provide a strong theoretical basis for VFDD, introducing the concept of Virtual-H divergence to explain why their method can overcome local optima issues. This theoretical insight adds depth to the proposed solution and helps understand its effectiveness. The presented illustrations and formulas clearly describe the current problems in DAOD, while providing a sufficiently detailed explanation and description of the proposed methods.
3.	The paper includes comprehensive experimental results across various benchmarks such as Cityscapes, Foggy Cityscapes, Sim10k, and BDD100k. These experiments demonstrate the robustness and general applicability of VFDD, showing consistent performance improvements over existing methods.

**Limitations:**

1.	In line 162, you mention part (b) of Figure 1, but your Figure 1 does not have a part (b). Are you referring to part (b) of Figure 2? If so, please correct this. Additionally, in Figure 2, you mention that the distributions of the source and target domains are similar, but I don't think this figure demonstrates the similarity in distributions. I suggest creating separate feature distribution plots for the source and target domains.
2.	The description of the VFDD technique in section 3.2 is too abstract and does not provide readers with a clear understanding. I suggest incorporating specific examples from the dataset to clarify:
(1)	What are V-Source (V-target) samples? Which samples in the dataset can serve as V-source (V-target) samples? Are there any boundary cases where samples cannot serve as V-source (V-target) samples?
(2)	For lines 395-401, I recommend using graphical illustrations with specific examples from the dataset to explain.
3.	The paper mentions the threshold τ as a crucial parameter for determining whether a sample is "unaligned" (Equation 9). However, the experimental section lacks a detailed sensitivity analysis of τ across different datasets and tasks. The threshold τ is briefly discussed in Figure 5(a) (Design Choices), but the exploration is limited to a narrow range of values. A more comprehensive analysis would involve varying τ across a wider range and reporting its effect on the performance metrics systematically.
4.	The paper provides a clear analysis of all the illustrations. For instance, the explanation of Figure 1 is clear and connects well with the narrative in the text, and Figure 3 is well-explained with step-by-step details, making the concept of VFDD understandable. However, some improvements can provide a clearer and more comprehensive understanding of the figures and their implications, thereby strengthening the overall narrative.
a)	For Figure 5, provide a more detailed interpretation of the trends observed in the influence of threshold τ and patience parameter K. Explain how different values impact the performance and why certain values might be optimal.
b)	For Figure 6, expand the explanation to discuss specific performance improvements of VFDD over RADA. Highlight any notable observations and potential reasons for the differences in performance.
5.	The scalability of VFDD to larger datasets or more challenging domain gaps is not thoroughly evaluated:
a)	Understanding how VFDD performs as the size and complexity of data and models increase is crucial for assessing its practical applicability in real-world scenarios. The paper does not include an in-depth analysis of scenarios where VFDD might fail or perform sub-optimally.
b)	In my opinion, there are particularly challenging scenarios of domain gaps in medical image analysis. Can the proposed method work effectively under larger domain gaps? For instance, the following articles [1]-[2] analyze the adaptation to significant domain gaps in medical scenarios.
[1] Shin, H., Kim, H., Kim, S., Jun, Y., Eo, T. and Hwang, D., 2023. SDC-UDA: volumetric unsupervised domain adaptation framework for slice-direction continuous cross-modality medical image segmentation. In Proceedings of the IEEE/CVF Conference on Computer Vision and Pattern Recognition (pp. 7412-7421).
[2] Fan, J., Liu, D., Chang, H., Huang, H., Chen, M. and Cai, W., 2023. Taxonomy adaptive cross-domain adaptation in medical imaging via optimization trajectory distillation. In Proceedings of the IEEE/CVF International Conference on Computer Vision (pp. 21174-21184).

**Suitability:**

2

---

### Official Review · Reviewer_jYWF · 2024-06-06

**Rating:** 4
**Confidence:** 2

**Summary:**

The paper aims to prevent the  domain classifier from being trapped in local optima by introducing virtual samples. The author also  provide a theoretical analysis of the proposed Virtual–H-divergence that it has a smaller upper-bound than the standard H-divergence.

**Strengths:**

1. The paper is well-written and easy to follow. The motivation is well-expressed with some preliminary experiments.
2. Theoretical proof is also provided to validate the proposed method.
3. Extensive experiments are provided.

**Limitations:**

1.The methods somehow share the same spirit as existing methods.[1] Although the problem is different, the high-level idea involves creating virtual samples based on current target and source samples to augment the dataset and achieve better accuracy.
2. The improvement in performance using the proposed methods does not seem significant.
Minor:
1. In line 165, the authors may refer to the wrong figure.
2. The captions of the figures are too long, it would be better to shorten them.
[1]Pairwise adversarial training for unsupervised class-imbalanced domain adaptation

**Suitability:**

2

---

### Official Review · Reviewer_T3Uf · 2024-06-07

**Rating:** 4
**Confidence:** 2

**Summary:**

This paper proposes an optimization strategy called Virtual Label Fooled Domain Discrimination for improving Domain Adaptive Object Detection (DAOD) methods, typically which are using the domain adversarial training approach. This method aims to address the equilibrium challenge during the training of the feature extractor and the domain classifier. Specifically, the authors propose to identify the unaligned samples for each mini-batch and assign virtual labels to them in order to confuse the classifier. The authors claim by this approach, the discriminator will be more likely not to fall into suboptimal points during training, which will lead to an improved feature extractor and more aligned feature space for source domain and target domain. The authors evaluate their methods on several datasets and show better results than previous methods.

**Strengths:**

1. The motivation is interesting. The idea of using virtual labels to confuse the classifier to improve the feature extractor might have contribution to the community.
2. The authors demonstrate the effectiveness of the methods on several datasets and compare with a variety of previous methods, which can be supportive.

**Limitations:**

1. The novelty of the approach might be limited. Although effective, the methods are straightforward. It is similar to applying some data augmentation methods during the training session. One can think of some variations and comparisons from this perspective, which might impact the novelty of the proposed method.
2. Some parts of this method can be further explored. The authors mention that how to select the unaligned samples is not the focus of the paper, however, given the simplicity of the proposed methods, some more exploration on this part could also be beneficial.

**Suitability:**

2

---

### Meta-Review · Area_Chair_JHqg · 2024-07-08

**Recommendation:** Accept (Poster)
**Confidence:** 4

**Metareview:**

The paper receives consistent weak acceptance ratings from all the reviewers. The paper presents a valid and technically sound solution for the adversarial domain adaptation. However, two of the reviewers reveal the novelty concern about the method. I read into the paper and find the similar concern, e.g., when adversarial learning goes to a stable state, which usually means the source and target domain are sort of well aligned. It would be good in the final version, the authors can address this concern, e.g., by pivoting the domain label the continuous adaptation can further improve the alignment or classification performance.